# Consultation and Displacement in Large-Scale Agriculture Investment: Evidence from Oromia Region's Shashamane Rural District

Yideg Alemu *[ID] and Degefa Tolossa

College of Development Studies, Addis Ababa University, Addis Ababa P.O. Box 1176, Ethiopia
* Correspondence: yideg.alemu@aau.edu.et

**Abstract:** The Shashamane rural district was selected as a target area and corridor of large-scale agriculture investment (LSAI) to produce surplus agricultural products and ensure local development by the state and private (domestic and foreign) investors. Shalo–Melega private LSAI projects started operation in 2008 in the Shashamane rural district. This farm project comprises a crop production site, construction of a road, a crop storage facility, and developing irrigation in a total of about 24,710.51 acres of land along the central Rift Valley basin, for long-term leases. Little attention has been paid to how land ownership has changed and transaction transparency; how the community has been consulted; whether free, prior, and informed consent (FPIC) has been provided; and how local people have been displaced. This study sought to investigate the consultation process, land transaction transparency, the use of FPIC, and local community dis-placement as a result of LSAI in the Shashamane rural district. The study adopted multi-method qualitative and quantitative data collection tools including primary data, collected from a directly impacted population of 134 households, using systematic random sampling techniques; key and in-depth informant interviews; focus group discussions (FGD); and field visits. Through the use of qualitative and qualitative research paradigms, a systematic analysis was conducted. The result of the study shows that 86.6% of respondents (both interested and affected) expressed that both government and the proponents were not taking in account their concerns during the consultation processes. Lack of free, prior, and informed consent (FPIC) reduces local people's sense of recognition and status. Moreover, LSAI displaced the rural people from their area of settlement and farmland, triggered a shortage of communal grazing and forest resources. Additionally, nonequivalent and unsatisfactory mitigation and compensation methods highly triggers the negative impacts. As a result of manipulation and therapy used during the consultation process, we assert that the local community had less decision-making authority and that the risk to the farm was thereby increased. The government, investors, and local communities are three actors whose respective roles need to be strengthened and transparent. It is crucial to strengthen the implementation of customary land tenure rights to benefit local and indigenous people and civil society organizations (CSOs).

**Keywords:** land dispute; displacement; stakeholder consultation; large-scale agriculture investment; Oromia region; Ethiopia

## 1. Introduction

Large-scale agricultural investments (LSAI) have been shaping and changing Africa's agriculture and food system structure since 2008 [1–3]. Most frequently, a sizable area of the occupied land is situated in the rural parts inhabited by indigenous peoples and is viewed as a common resource [4]. Africa is a prime target of this development where populations are rapidly displaced and dislocated, as their prime lands are leased for agro-production meant for overseas economies [5]. The World Bank believes that in just one year, 2008–2009, there was a 14-times increase in the number of deals disclosed, despite disagreement over

the numbers [6]. The present "land rush" is notable for the speed with which the demand for land grew, starting in 2008, as well as the scale of the purchases, the long-term nature of the agreements, and the global reach of the phenomena [7]. These types of agreements used in the last wave of land acquisitions typically involve long leases, lasting between 50 and 99 years, and involve purchases of more than 10,000 hectares [8]. More than a dozen African nations, including Ethiopia, have distributed millions of hectares of farmland to investors as of the end of 2009 in the hope that LSAI would open opportunities for quick agrarian development and act as a key tool for addressing enduring rural poverty [9]. LSAI is also referred to as land grabbing, new land colonization, or green colonization [6,10,11].

Policymakers, academics, the media (private and public), and NGOs (nongovernmental organizations) from civil society continue to debate the contribution of LSAI. Additionally, the fundamental justification for large-scale agricultural investments in Africa has been the likelihood that they would create jobs for the local population [12,13]. Studies have discovered benefits for employment and rural welfare [14]. Investors frequently claim that these land purchases would increase local employment and advance technology to increase crop yields, but these advantages rarely manifest [13,15]. However, large-scale agricultural investments have a variety of different effects on the environment, food security, and livelihoods in their target regions [1,16–20]. Other research demonstrated that LSAIs uproot land users, weaken resilience, dismantle institutions of traditional land tenure, and cause destruction of livelihoods, deforestation, environmental degradation, and increase conflict [21–23]. As a result, the socioeconomic, food security, and environmental effects of LSAIs vary greatly depending on the environment [24].

Despite the increase in investment in rural areas, locals are worried about the possible negative effects on their quality of life, including access to agricultural land, productivity, income levels, food security, and access to social services [14,25–28]. Most LSAI research has focused on the wellbeing effects, with an emphasis on increased income, employment possibilities, the development of new management and agricultural skills, the transfer of technology, agricultural productivity, food security, and access to natural resources. Although widely acknowledged as a significant factor and influencer of land agreements, stakeholder consultation processes; free, prior, and informed consent (FPIC); and displacement remain undefined in terms of concrete research [29–31]. The results of these substantial investments depend on how they are carried out [32]. The business models in place are the culmination of all the development initiatives defining the type of linkages, partnerships, and relations, along with the legal environment in the investor's country of origin, investor–community linkages, and the type of partnership with the governments—all influence the implementation models, which vary both within and between communities [33].

Moreover, the actual use of FPIC, however, frequently falls far short of the ideal [34–39]. Based on the principle of Free, Prior and Informed Consent (FPIC) respects indigenous peoples' legitimate right to demand that third parties treat them with equality and respect. Procedurally, free, prior, and informed consent requires processes that enable and support meaningful decisions about indigenous peoples' development paths [40]. All people's right to full consultation, expression of views, and compensation, including relocation with adequate state assistance, participation in national development, and, in particular, consultation concerning policies and projects affecting their community, together with improvement of their capacities for development and to meet their basic needs, are explicitly recognized in Articles [41–44], and 92 of the Constitution of the FDRE [35]. Public participation is a necessary legal requirement for the implementation of significant development projects, programs, and plans, according to the Environmental Impact Assessment Proclamation (Proc. no. 299/2002) [36] and Environmental Policy of Ethiopia [43]. This declaration and policy served as a proactive instrument and the foundation for integrating environmental, economic, cultural, and social factors into decision-making in a way that supports sustainable development. Moreover, large-scale agricultural production initiatives, which include planting, transplanting, growing, and collecting plant material, are acknowledged as requiring stockholder and public consultation [44]. However, there

are significant normative gaps in the law, community consultation, and actual practice surrounding these rights.

However, there are large normative gaps about these rights in the law, community consultation, and actual practice [45]. For instance, the recent Oromo Protests (2014–2017) were sparked by land expropriations and the removal of forest areas without proper compensation in the Oromia Region [46] The Addis Ababa Integrated Regional Development Plan was brought about by the Federal Government [40]. Millions of Oromo smallholders in the Special Zone of Oromia Surrounding Addis Ababa (Finfinnee) are believed to have been uprooted from their farms as a result of the Master Plan's implementation, without proper compensation or resettlement options [45,46]. Further, the implementation of FPIC, stakeholder consultation processes, local affected and interested parties' perspectives, local people's displacement, and the interdependencies of LSLAs with the local community are all generally unknown in Ethiopia, and the Oromia regional state in particular [45,47].

This study presents recent empirical research on the consultation process, application of FPIC, and background of current LSAI to investigate the level of involvement of local community households in deal-making and compensation. In fact, this study sought to answer four research questions and contribute to the debate on LSAI. This involved bringing the effects of LSAI to government agenda and intervention, and by facilitating monitoring and evaluation of projects and institutions whose goal is to monitor and evaluate LSAI, such as the federal and regional land administration and land use and lease, investment and environmental protection, and agriculture and rural development offices. To begin, do Shalo–Melega LSAIs include affected and interested parties' perspectives, and is FPIC used? Second, are appropriate augmentation and mitigating actions put in place as soon as possible, ideally during project design and execution? Third, what is the government's relationship with the local community and investors? Fourth, how do perception and participation in the community influence LSAI? The remainder of this article is structured as follows: the summary of the research technique and approach is given in Section 2; the framework for conceptually examining the degree of local community participation in LSAI and consultation outcome is described in Section 3; the results are given and discussed in Sections 4 and 5; and the final section includes conclusions and perspectives.

## 2. Research Methodology and Research Approaches

### 2.1. Choosing a Research Location

Five factors were taken into consideration when the research location Shashamane rural district was chosen for this study:

1. There has been or is currently an LSAI process.
2. The Shalo–Melega LSAI in Ethiopia's Shashamane area of Oromia experienced low community engagement and application of FIPC.
3. Shashamane rural district is perhaps the most known for its higher demographic pressure and land shortage.
4. The local and indigenous population primarily engages in smallholder agriculture and natural resource.

### 2.2. Background Overview Study Area and the Project

The Shashamane Rural District is located topographically in the West Arsi Zone of the Central Main Ethiopian Rift Valley. The Shashamane Rural district is bordered to the north by the Arsi Ngela district, to the west by Bishan Guracha Town, and to the south by the Wndo district of SNNP. A commercial hub called Shashamane Town is roughly 240 km (150 miles) from Addis Ababa, the country's capital. Most of the population comprises Oromo smallholder farmers who rely on communal lands and local resources. A total of 28 kebeles (sub districts) make up the district, which has a population of 125,000 people overall. The land cover of the district consists of arable land, open woodland, grazing land, woodland, and shrub land. The principal crops farmed—their primary economic activity, smallholder farming—include wheat, sorghum, maize, teff, oil seeds, and spices.

There is a high population of livestock, and raising livestock is a significant source of revenue. Since 1993, Mohammed International Development Research and Organization Companies (MIDROC) have been operating in the private sector of Ethiopia's economy. The groups were able to expand rapidly, largely due to their humble beginnings. Their presence has a significant impact on the country's economy, as their large investments and many different activities make a significant impact [48]. With the help of four independently operating business groups, MIDROC Ethiopia has been successful in creating a sizable local corporate empire. With more than 70 PLCs (private limited companies) owned by the Investment, Technology, Horizon, and Derba groups, MIDROC Ethiopia's diverse groups are the greatest private economic empire in the nation. One of the MIDROC Ethiopia Investment Group Companies, Elfora Agro-Industries P.L.C. Shalo–Melga LSAI agricultural project began operations in 2008 and is located on 10,000 hectares in the nearby district of Shashamane in the Oromia region of Ethiopia. Elfora Agro-Industries is a company that produces agricultural products. The Shalo-Melega farm currently grows commercial maize (*BH661*, *BH546*), wheat, haricot beans (*Nassri*, *Awassa Dume*), and white beans, and soybeans for domestic and international markets. The area is located near the large farm of Elfora Agro-Industries P.L.C, about 7 and 5 km east and west, respectively. Looking at the background of the land relocated to the Shalo–Melega district, it was once owned by rural households in rural areas, with cultivation in villages, and was regarded as general pasture land. A variety of food crops were grown until the land was transferred to Elfora Agro-Industries P.L.C's large-scale crop production. Additionally, the region offered animals and humans access to water supplies, beneficial vegetation, and firewood. For shareholder consultation, both rain-fed and irrigated food crops are taken into account. This production typically involves sophisticated food production methods that use agricultural inputs. Irrigation plans are used to grow irrigated crops, which boosts agricultural output and farmer income. Food crops for industrial/export production are raised in both highlands and lowlands, in a range of agro-ecological zones. Industrial/export crops farmed on a large scale in Elfora Shalo–Melega are irrigated and rain-fed. Crop cultivation is often intense and protective measures are used for industrial and export crops. Large monocultures are the norm for agricultural crop production, and heavy equipment is frequently used in modern farming practices for plowing, sowing, and harvesting; fertilizer and herbicide application; and irrigation systems. Crop production initiatives can be an element of the watershed and integrated rural development programs.

### 2.3. Research Design

A mixed or combined (qualitative and quantitative) paradigm of research was used for this study. Mixed methods research aims to justify the use of many approaches to resolving research problems rather than limiting or restricting researchers' options (i.e., it rejects dogmatism); it also is an open-minded and inventive method of inquiry rather than one that is restricted. Being inclusive, pluralistic, and complementary enables researchers to embrace a diversified approach to method selection, research planning, and actual research [49]. The research question is the most important factor; research methods should be chosen in a way that many research questions and combinations of questions are best and most fully answered through mixed research solutions [49,50]. In fact, this research requires in-depth understanding of the given phenomena, representing an attempt to provide warranted assertions about human beings (or specific groups of human beings) and the environments in which they live and evolve [49].

### 2.3.1. Data and Sampling

In Ethiopia's Oromia state's Shashamane district, primary data were gathered through household surveys. The Elfora Shalo–Melega LSAI is held in these districts (Figure 1). Elfora large-scale crop production is a member of MDROC Group, which was founded in 2008 on 10,000 ha of land in Ethiopia's Shashamane District of the Oromia regional state. These LSAIs, which were constructed with little input from residents and stake-

holders, support the eviction of smallholders and restrict access to grazing during the dry season. Out of the seven closest kebeles, three were chosen at random, and these three kebeles are close to the LSAI and can be located within a ten kilometer radius of the LSLI. First, the Shahsemena rural district was specifically chosen for the existence of the LSLI. Primary data were collected through a household survey in the Shashamane district located in the Oromia state of Ethiopia. These districts host the Elfora Shalo–Melega LSAI (Figure 1). Second, the district health office's numbers from two months prior estimate that 2098 people are living in the three kebeles as a whole (1784 male and 314 female). Furthermore, approximately 85% of the sample population in this study engaged in mixed agriculture (i.e., farming and livestock). Third, the sample size was calculated using the Cochran, (1977) and Robert (1986) formula, which takes into account a 90% confidence level (z = 1.64), a 70% estimated proportion of a character in the population (p), and a 7% level of precision (E). Finally, 134 families from a population that was directly or indirectly affected were chosen at random based on likelihood proportionality to sample size. Additionally, long-term residents in the area, 18 years or older, experienced with LSAI have been used as including criteria. Conversely, for this investigation, we set a precision level of 7% while taking into account the resources that may be employed to manage the study.

$$n = \frac{Z^2 P q}{e^2} = \frac{1.64^2 * 0.5(1 - 0.5)}{0.07^2} = \frac{2.6896 * 0.5 * 0.5}{0.0049} = 137.22 \sim 138$$

where:

$n_o$ is the sample size;

$Z$ is the selected critical value of desired confidence level;

$P$ is degree of variability in the population;

$q = 1 - P$ and $E$ is the desired level of precision.

$$n = \frac{n_o}{1 + \frac{(n_o - 1)}{N}} = n = \frac{138}{1 + \frac{(138 - 1)}{2098}} = 130.\,28 \sim 134$$

where = $n$ is the desired sample size,

$$ni = \frac{n * Ni}{N}$$

where ni = sample of kebeles, $Ni$ = population of kebele, $n$ = total sample size, and $N$ is the total population of the three kebeles. For the sample drawn from each kebele, see Appendix A, Table A1.

### 2.3.2. Data Collection Tools

The effects of LSAI on eviction, displacement, and compensation (mitigation) mechanisms were investigated using a multi-method qualitative methodology that included household surveys, key informant interviews (KIF), focus group discussions (FGDs), field visits, and observation. KIT and FGD were guided by a 32-item checklist of COREQ (consolidated criteria for reporting qualitative studies principles [51]. Further, Table 1 identifies the variables under investigation.

A.  Questionnaire survey

The study employed closed-ended questions to ask about households' socioeconomic status, tenure structures, and access to land. Further, information and data related to community participation, consultative process, land confiscation, and forcible relocation of locals' community were asked to rate their overall satisfaction with the project on a 4-point scale (0 being not at all satisfied, 1 being poor, 2 being medium, and 3 being very satisfied), and the degree of participation and consultation process as depicted in the conceptual formwork (see conceptual formwork of the study). They were then asked for any action the government had done to monitor and assess the LSAI's performance. By interrogating

respondents and asking them to defend their complaints, efforts were taken to reduce any bias. Additionally, a pilot survey designed to catch misconceptions in the questions was used to pretest and check the questionnaires. The survey's primary goal was to collect common characteristics, effects and mitigation, opinions or beliefs, and experiences of local communities currently residing in LSAI areas. A trained enumerator of development agent (DA) then distributed the questionnaire to the respondents in each kebele. With the help of the district agricultural experts, the investigator, and a total of six enumerators, respondents were interviewed door to door; 134 (100%) of households that took part in the survey responded quickly to the questionnaire. In this study, Cronbach's alpha, a bias indicator of the questionnaire's internal consistency and scale reliability, fell within an acceptable range [52].

B.     In-depth conversations

One-on-one, in-depth conversations with respondents were a part of the interviews. A total of 35 households were interviewed: 15 were from the B/Dannaba, 10 were from the Toga, and 8 were from the D/Calalaqaa kebele. Interviewers were selected based their experience, academic credentials, and ability to communicate in the local language (Afaan Oromo). Moreover, training was given to all interviewers. The main questions during an interview concerned LSAI phenomena, perceptions, and experiences in the last ten years. These questions concerned consultation, relocation, mitigation, and rehabilitation. Investigations focused on factors that contribute to displacement, such as LSAI and development initiatives that were mentioned in the conceptual framework (Figure 1). Separate interviews with identifiable codes for each participant took place in a quiet setting, either at home or in a local community center. When the thematic saturation was reached, interviews were terminated. A notebook was used to record the interview. The length of in-depth interviews was 20 to 25 min.

C.     Interviewing key informants

Additionally, 28 government employees (purposively selected sample respondents), who are in charge of overseeing large-scale farms and are employed by various federal and regional offices, were interviewed. This included specialists from the district council; land administration offices, agriculture and rural development offices; regional and district investment offices; Elfora Shalo–Melega LSAI (organization managers, experts, and administrators); and the Ethiopian investment agency.

D.     Focus Group Conversations (FGDs)

The research used six focus group conversations in two phases, i.e., two per kebele. However, people involved in the FGD varied across kebeles and phases. For instance, the number of people involved in the Toga kebele FGD during the first and second phase was 6 and 7, respectively. The number of people involved in the B/Dannaba kebele FGD during the first and second phase was 10 and 6, respectively. The number of people involved in the D/Calalaqaa kebele FGD during the first and second phase was 7 and 6, respectively. Moreover, this data collection tool and guide enabled and improved the quality of qualitative data, as previously mentioned. Additionally, transect walks and personal observation of the institutions and farm sites, and community resource mapping were employed. To illustrate or emphasize a particular issue, two images are provided in this report (Figures 2 and 3).

*2.4. Data Analysis*

Descriptive statistics used to analyze the data, are reported in tables. To analyze the data, descriptive statistics are used, as well as frequency and percentages. To increase a broader and deeper understanding of the research phenomenon and to improve the validity of the results, triangulation is used because it is more precise and because it aims to reveal complementarities, convergences, and inconsistencies within the research results [46].

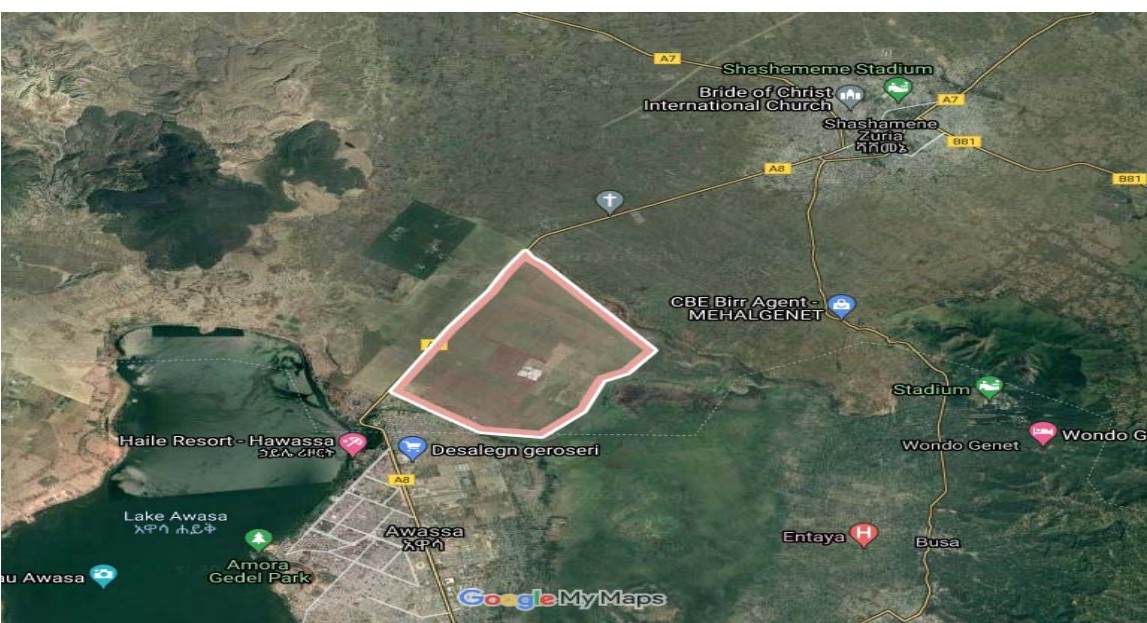

**Figure 1.** Rain-fed and irrigated crop production of the Elfora Shalo–Melga farm. The farm grows a variety of crops, including maize, sorghum, and soybeans. Food crops include grains, such as corn, wheat, rice, and other types of crops (e.g., vegetables and fruits). (Source: Google Maps).

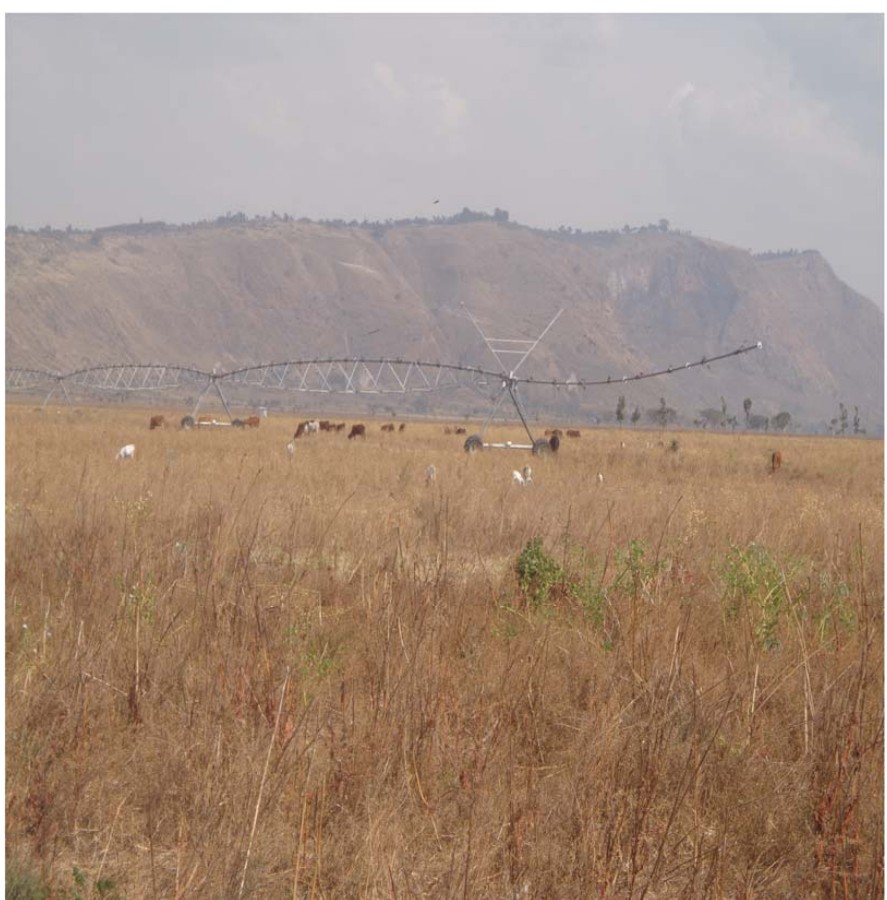

**Figure 2.** Elfora large-scale crop production farm in the Shashamane district, which is responsible for planting, transplanting, growing, and harvesting plant material, including food crops and export/industrial crops (right side). (Source: authors).

**Table 1.** List of major variables investigated.

| Dimension | Variable Name | Variable Type | Unit of Measurement |
|---|---|---|---|
| Recognition and sense of ownership | Meetings | Dummy | 1, if invited to a stakeholder meeting, 0 if not |
| | Participation | Dummy | 1, if participating in the proposal, 0 if not |
| | Proposal explanation | Dummy | 1, if the proposal is adequately explained to all stakeholders, 0 if not |
| | Effects and hazards | Dummy | 1, if the proposal effect and hazards are adequately explained, 0 if not |
| Power and level of influence | Responsibility | Dummy | 1, if the local community is influenced and empowered, 0 if not |
| | Decisions | Dummy | 1, if the decision of the local community incorporated, 0 if not |
| | Degree of community control | Dummy | 1, if higher degree of community control or partnership, 0 if not |
| Key principle of participation | Open and transparent | Dummy | 1, if participation is open and transparent, and understood, 0 if not |
| | Fair and neutral | Dummy | 1 if, the consultation is fair and neural, 0 if not |
| | Inclusive | Dummy | 1, if the proposal is inclusive, 0 if not |
| | Relevant | Dummy | 1, if the proposal is, 0 if not |
| | Responsive | Dummy | 1, if the proposal is responsive to stakeholder input, 0 if not |
| | Credible | Dummy | 1, if the proposal is credible, 0 if otherwise |
| Impact | Direct impact | Dummy | 1, if the proposal is a direct impact, 0 if not |
| | Displacement from resident | Dummy | 1, if the proposal is to displace you from your residential, 0 if not |
| | Displacement from communal land | Dummy | 1, if the displace your from communal land, 0 if not |
| | Indirect impact | Dummy | 1, if the proposal is an indirect impact, 0 if not |
| | Cultural sites | Dummy | 1, if the proposal is adequately explained, 0 if not |
| | Heritage | Dummy | 1, if the proposal is affecting heritage explained, 0 if not |
| | Biodiversity | Dummy | 1, if the proposal affects biodiversity, 0 if not |
| Mitigation and compensation | In-kind compensation | Dummy | 1, if in-kind compensation is provided, 0 if otherwise |
| | Monetary compensation | Dummy | 1, if the monetary compensation is provided, 0 if not |
| | Resettlement | Dummy | 1, if compensation is provided resettlement, 0 if otherwise |
| | Site remediation | Dummy | 1, if Site remediation is provided, 0 if otherwise |
| Mentoring | Monitoring and follow-up | Dummy | 1, if the monitoring and follow-up are adequate, 0 if not |
| Satisfaction | Satisfaction | Rank (4-point scale) | 0 being not at all satisfied, 1 being poor satisfaction, 2 being medium satisfaction, and 3 being very satisfied) |

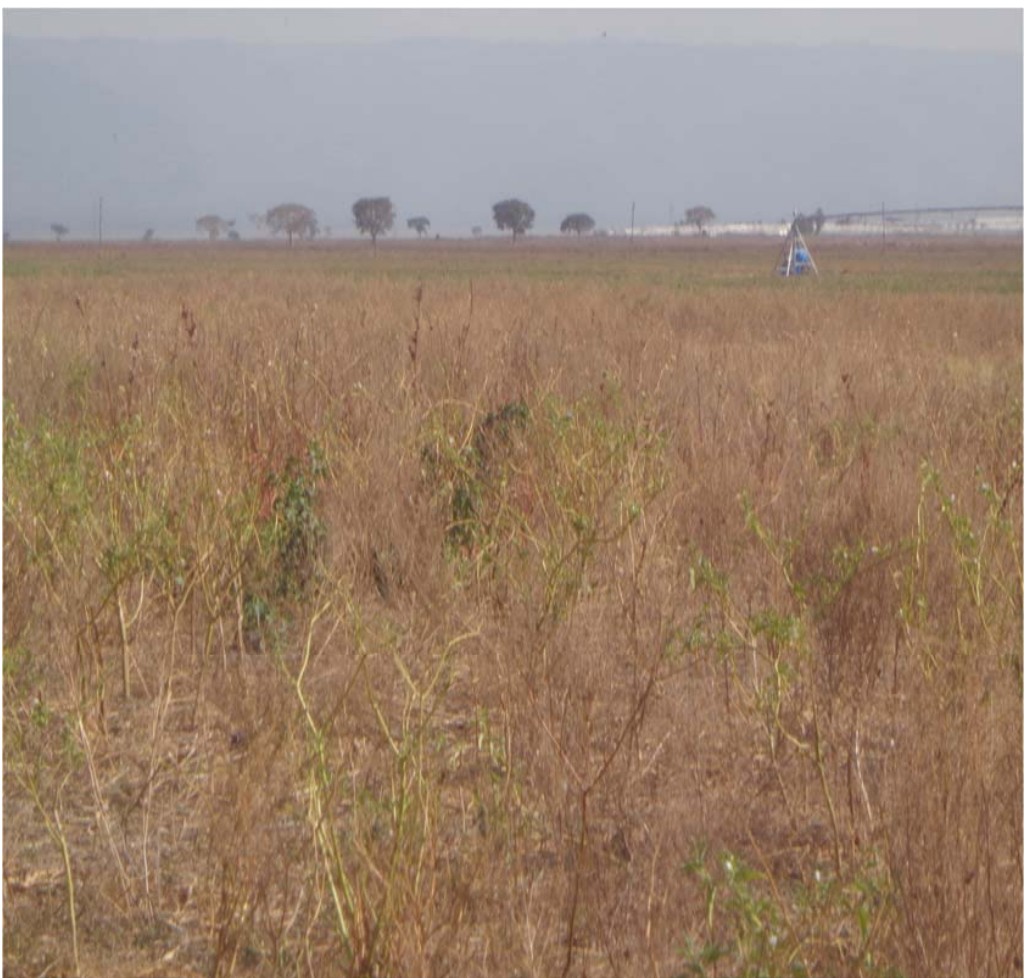

**Figure 3.** Elfora large-scale crop production farm in the Shashamane district, which is responsible for planting, transplanting, growing, and harvesting plant material, including food crops and export/industrial crops (left side). (Source: authors).

### 3. Conceptual Framework for Analyzing Local People's Community Participation in LSAI

Figure 4 shows our conceptual framework used to analyze local people's community participation in LSAI. It draws on Arnstein's and Guaraldo's concept of ladders of participation and degrees of citizen power [53,54]. At the bottom of the ladder, the community has almost no decision-making power. Moving up the ladder, the community exerts increasing influence, to the point where they make the decision at the top. The terms "community involvement" and "participation" are used synonymously in this study to refer to the involvement or participation of the community of households in the creation and implementation of initiatives and programs that affect them, along with the formal decision-making process. Public involvement contributes to a project's success and sense of ownership over the medium to long term [55]; some are shown in Figure 4. The primary goal of public participation is to motivate the populace to contribute meaningfully to the decision-making process [56]. Many people enjoyed being consulted because it gives them a sense of recognition and status. The main objective of public participation is to inspire the public to actively participate in decision-making [50]. These advantages occur when public involvement is a two-way process, allowing the agency and the public to both learn and profit [54–57]. The identification of the public's values and their implementation into decisions that eventually affect them are made possible through effective public engagement [52]. Hence, community perception and participation determine the sense of ownership, success, and sustainability of a project (such as LSAI). However, some

argue that involving the local community in a project was time-consuming, and including the view and interests of locals is very difficult. In Ethiopia, community participation in agricultural investment projects is generally not always large, particularly in investment regions such as Oromia regional state, and the Shashamane rural district, due to differences in community characteristics and phenomena that can also affect the level of communityperception and participation.

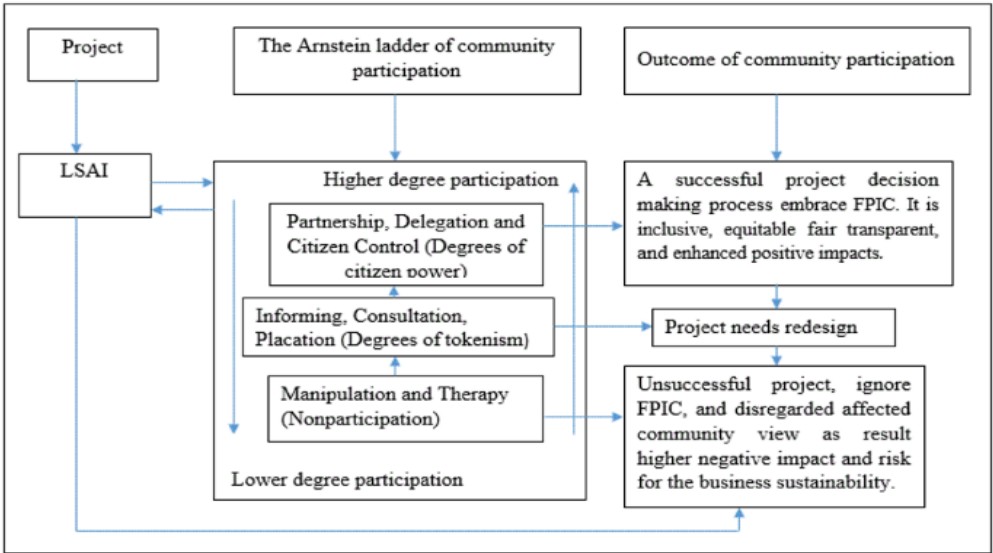

**Figure 4.** A conceptual framework for analyzing local people consultation and participation in LSAI. Source: authors' compilation developed from Arnstein and Guaraldo [47,53].

### 4. Results

The result part is divided into six sections. The first part shows the consultation process and FPIC of local people in LSAI in the Shashamane rural district of Oromia regional state. The second part covers the direct and indirect effects of the LSAI proposal on the wellbeing of local communities and the success of investment projects. The third and fourth parts present linking to the participant's basic understanding of the effects of LSAI on unplanned human settlements, social conflicts due to shortage of settlement and communal grazing land, the venue of nonresident workers and migrants, and water disputes, and the consequences on the society. The fifth part includes enhancement and mitigation measures, including providing settlement areas with appropriate housing and social services (water, school, and sanitation) for displaced families and nonresident workers and their families. Finally, based on the data gathered during the field study, the overall satisfaction of the local community with LSAI is presented.

### 4.1. Stakeholder Consultation and FPIC of Local People in the Shashamane Rural District LSAI

Table 2 depicts the views of affected and interested stakeholders on the LSAI and its effect. Approximately 86.6% of respondents express that their concerns were not taken into account and consultation processes were irresponsive to stakeholder requirements and inputs; 13.4% of the respondents were neither consulted nor informed about the LSAI. Moreover, findings from the consultation are inconclusive, unfair, and not open and transparent, lacking credibility that builds confidence and trust between the government, the proponent, and the local community in the study area. People directly impacted by the plan must, at the very least, have the opportunity to express their opinions about it and any potential social, environmental, or other repercussions.

**Table 2.** Involvement during the consultation application of FPIC or two-way exchange of information (*n = 134*).

| Consultation and FPIC Component | | Frequency | Percentage (%) |
|---|---|---|---|
| Stakeholder meetings | Yes = 1 | 116 | 86.6 |
| | No = 0 | 18 | 13.4 |
| Proposal explanation | Yes = 1 | 10 | 7.4 |
| | No = 0 | 124 | 92.5 |
| Potential effects and hazards | Yes = 1 | 8 | 6 |
| | No = 0 | 126 | 94 |
| Power through and a sense of social responsibility | Yes = 1 | 4 | 2.1 |
| | No = 0 | 130 | 97.9 |
| Decisions are made and based | Yes = 1 | 7 | 5.2 |
| | No = 0 | 127 | 94.8 |
| Voice issues and impact on the decision-making process | Yes = 1 | 3 | 2.2 |
| | No = 0 | 131 | 97.8 |
| Inclusive—covers all stakeholders | Yes = 1 | 6 | 4.5 |
| | No = 0 | 128 | 95.5 |
| Open and transparent? | Yes = 1 | 9 | 6.8 |
| | No = 0 | 125 | 93.2 |
| Fair, neutral, and performed without prejudice | Yes = 1 | 5 | 3.8 |
| | No = 0 | 129 | 96.2 |
| Responsive to stakeholder requirements and inputs | Yes = 1 | 1 | 0.7 |
| | No = 0 | 133 | 99.3 |
| Credible—builds confidence and trust | Yes = 1 | 2 | 1.5 |
| | No = 0 | 132 | 98.5 |
| Gender issue | Yes = 1 | 4 | 2.1 |
| | No = 0 | 130 | 97.9 |

*4.2. Direct and Indirect Effects of an LSAI Proposal*

LSAI was recognized for both its immediate and long-term effects. A direct loss can be measured in a specific way, such as the number of locals who were displaced or the amount of property, infrastructure, and natural resources that were harmed. More vulnerable are those who are poor, landless, tend livestock, have a big family size, and are women or elderly men. The powerful can do what they please with the poor. Indirect losses typically emerge from disruptions to the flow of products and services brought on by large-scale crop production initiatives and include drops in output or revenue along with effects on people's wellbeing. Based on the result of the survey, 97.9% of rural households in the three kebeles (B/Dannaba Toga and D/Calalaqaa) were directly or indirectly affected by the LSAI proposal (see Table 3). Based on the survey result, 20.9% of rural households were displaced from their locality (dwelling) or residential homes. According to the Shashamane rural district investment office, interviewed for this study, the Shalo–Melga LSAI project has displaced 2980 individuals, and the local community is concerned about further evictions due to the government and company's expunction to new arable lands. Almost all kebele administration leaders and elders from B/Dannaba Toga and D/Calalaqaa interviewed thought the LSAI has caused displacement and resulted in conflict between local peasants and investors.

**Table 3.** Local people affected by the proposal *n = 134*.

| Effect of LSAI | | Frequency | Percentage (%) |
|---|---|---|---|
| Local and indigenous people are affected by a proposal? | Yes = 1 | 130 | 97.9 |
| | No = 0 | 4 | 2.1 |

Moreover, 52.9% and 91.7% of rural households were displaced from their farmland and communal grazing land without adequate compensation for losses. In all three sites we studied, there was a drastic change in this regard. The district agricultural office trained

farmers to reduce the number of livestock and improve and use their private enclosure. Government extension agents tried to persuade farmers to start private enclosures where they did not exist. According to the district animal scientist, the absence of grazing land and forestland has a serious problem affecting the local community in the district. Moreover, he also emphasizes displacement from the locality and occupational activities, and loss of farmland and grazing land were common in the study district.

In the explorative study, the focus group discussant (FGD) explained that:

*"LSAI are causing large-scale displacement and communities are at great risk of mass dispossession today young people have no alternative and we have not enough land to share with them. Migration to the urban area, Shashemena town, Aris Negela, and Addis Ababa out of Ethiopia to Arab countries such as Saudi Arabia, Kenya, and South Africa, is the only alternative to minimize the household pressure, at least they feed themselves and send some money to for family."*

(FGD interviewed, D/Calalaqaa kebele, 2020)

Loss of cultural, religious, and historical heritage assets, together with the loss of aesthetic resources, are other significant problems that could develop when building and/or operating a rain-fed and irrigation agriculture production project. In this regard, our survey results reveal that 91% and 86.5% of rural households indicated that the LSAI affects cultural sites: religious and historical heritage assets and aesthetic resources, respectively (see Table 4).

**Table 4.** Local people displacement due to the LSAI.

| Displacement and Mitigation Measures | | Frequency | Percentage (%) |
|---|---|---|---|
| Displacement from locality | Yes = 1 | 28 | 20.9 |
| | No = 0 | 106 | 79.1 |
| Displacement from farmland | Yes = 1 | 71 | 52.9 |
| | No = 0 | 63 | 147.1 |
| Displacement on local grazing land | Yes = 1 | 128 | 95.5 |
| | No = 0 | 6 | 4.5 |
| Loss of cultural, religious, and historical heritage assets | Yes = 1 | 123 | 91.7 |
| | No = 0 | 11 | 8.3 |
| Loss of aesthetic resources | Yes = 1 | 116 | 86.5 |
| | No = 0 | 20 | 13.5 |

*4.3. Causal Association between LSAI and Human Health Outcomes, Loss of Crop Production, and Unplanned Human Settlements*

Table 5 presents the results linking to the participants' basic understanding of the effects of LSAI on human health-related outcomes, loss of crop production, unplanned human settlements, and social conflicts due to shortage of settlement and grazing land, the venue of nonresident workers and migrants, and water disputes. We observed that land acquisitions were a major source of social tension in the district. Many disputes and conflicts arise over land compensations between local governments and farmers. Greater than 75.4%, 76.8%, 73.1%, and 60.5% recognized a causal association between LSAI and human health outcomes, loss of crop production, unplanned human settlements, and social conflicts due to shortage of settlement area, communal grazing land, venue of nonresident workers and migrants, and water disputes, respectively. A dairy farmer from Shashamane rural district said *"LSAI affects all of us . . . because our life is dependent on subsistence agriculture and animal rearing, because of disruption of communal land and lack of farmland was leading to loss on crop and dairy product and further it intensify social conflicts between farmer and investors."*

**Table 5.** Basic understanding of the effects of LSAI on human health-related outcomes, loss of crop production, unplanned human settlements, and social conflicts.

| Please Answer the Following Questions | | Frequency | Percentage (%) |
|---|---|---|---|
| Communicable diseases such as malaria | Yes = 1 | 101 | 75.4 |
| | No = 0 | 33 | 24.6 |
| Loss of crops | Yes = 1 | 103 | 76.8 |
| | No = 0 | 31 | 23.2 |
| Unplanned human settlements | Yes = 1 | 98 | 73.1 |
| | No = 0 | 36 | 26.8 |
| Conflicts | Yes = 1 | 81 | 60.5 |
| | No = 0 | 53 | 39.5 |

Another key informant also illustrated:

*"I have faced the constraint of livestock grazing, due to this I have enforced to sell livestock with cheap price or keep in the house without animal fodder and this LSAI also caused a shortage of traditional energy source to obtain from the forests and timber products".*

(Resident, interviewed, B/Dannaba 2020)

The land is a crucial resource since, according to the Shashamane Rural District Agricultural Office Development Agent, over 83 percent of the district's population relies nearly completely on agriculture for their living. Owing to the LSAI, there is not enough arable land to provide all agricultural needs for subsistence. Make sure that the underprivileged and other disadvantaged groups continue to have access to nearby, productive land for growing their food or for pasture.

Another key informant also illustrated:

I have seen very weak management of water resources and pesticide/insecticide storage (appropriate containers, and locked facilities), which is leading to exposing the local community spill overt effects of pesticides, other harmful chemicals, and communicable diseases such as malaria, and diarrhea.

In the explorative study, the focus group discussant (FGD) explained that:

*"Loss of vegetation and vital natural resources due to land clearing, loss of forest products (fuel wood, timber, nontimber forest products) have become high-priced, especially, resource for livestock production remains limited over time to time."*

(FGD interviewed, Toga kebele, 2020)

*4.4. Enhancement and Mitigation Measures for Settlement Areas*

The goal of mitigation is to find ways to protect the area that the plan will affect. It identifies the most effective techniques for mitigating, preventing, and reversing effects. The provision of fair compensation for farmers whose land has been expropriated is one of the major difficulties associated with rural land acquisition. Concerning this, our survey result reveals that out of a total of 28 displaced from their settlement area, 23 (82.14%), 2 (7.17%), and 3 (10.71%) of the rural households were mitigated with a resettlement package (Figure 5). However, no one was compensated for a performance bond, insurance, or bank guarantee. Respondents were asked whether or not compensation was adequate; all of the respondents indicated that it was adequate and promised at the time of consultation, but compensation was not provided as promised. The project was established in adequate resettlement areas without appropriate housing and services (water and sanitation).

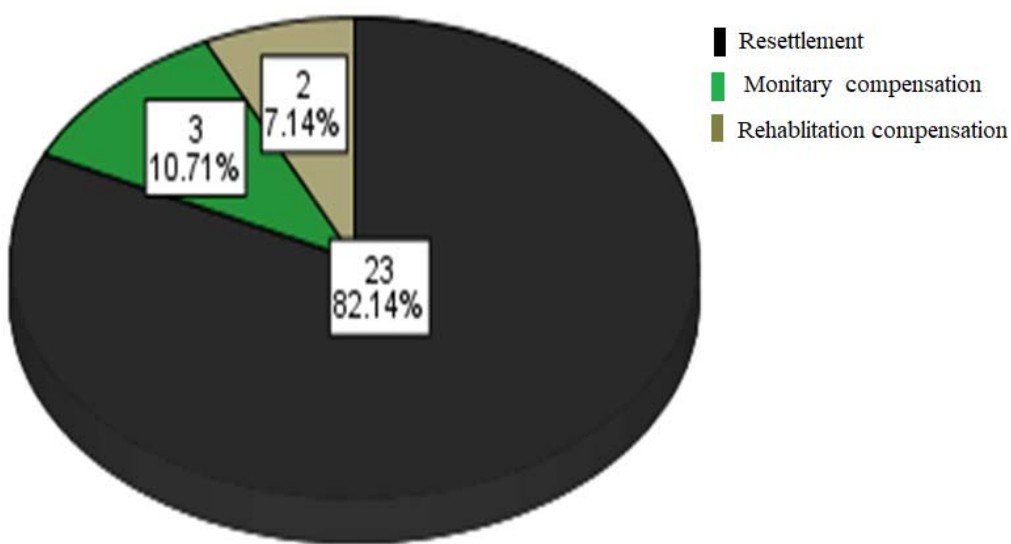

**Figure 5.** Enhancement and mitigation measures in settlement areas.

The key informant involved in Toga kebele added:

*"LSAI is typically cultivated in sizable monocultures, and contemporary farming techniques frequently require the use of fertilizers and pesticides, irrigation systems, and heavy machinery for plowing, planting, and harvesting, which causes disruption of agricultural habits in the local community."* In addition to displacement from settlement areas or locality or environment and occupational activities without adequate compensation, the local community in the three kebeles was losing farmland without adequate compensation. Of the total 78 displaced from their farmland, 60 (84.51%), 9 (12.66%), and 2 (8.2%) of the rural households were given replacement farmland at another location, provided monetary compensation, or offered rehabilitation of existing farmland, respectively, although none received payment for a performance bond, insurance, or bank guarantee. However, the compensations due to the resources offered to local community protest agents as compensation at another place were not equal. In addition, experts from the Ethiopian Investment Agency, regional and district investment offices, the district council, land administration offices, and agriculture and rural development offices confirmed that the necessary enhancement and mitigation measures were not included as early as possible, ideally during the project design. Moreover, about 55% were unaware of any action taken by the government to mitigate traditional cultural values, spiritual assets, and tourist attractions (Figure 6).

*4.5. Tracking Performance and Final Remarks/Comments from the Participants*

To determine whose rights can be upheld or what compromises can be reached, numerous communities have pleaded with their governments to step in when dispossession has happened in various regions of the world. Our survey result indicated that 65.6% of respondents appealed to the local government to monitor and evaluate the performance of the LSAI, while 93% of respondents expressed willingness to return the compensation if the government permits return of the lands previously owned (Appendix B, Table A2).

*4.6. Local Community's Satisfaction with the Overall Project*

The local community was asked to rate their overall satisfaction with the project on a 4-point scale (0 being not at all satisfied, 1 being poor, 3 being medium, and 4 being very satisfied). Of the 134 household respondents, 94%were unsatisfied by the project activity (Appendix C, Figure A1).

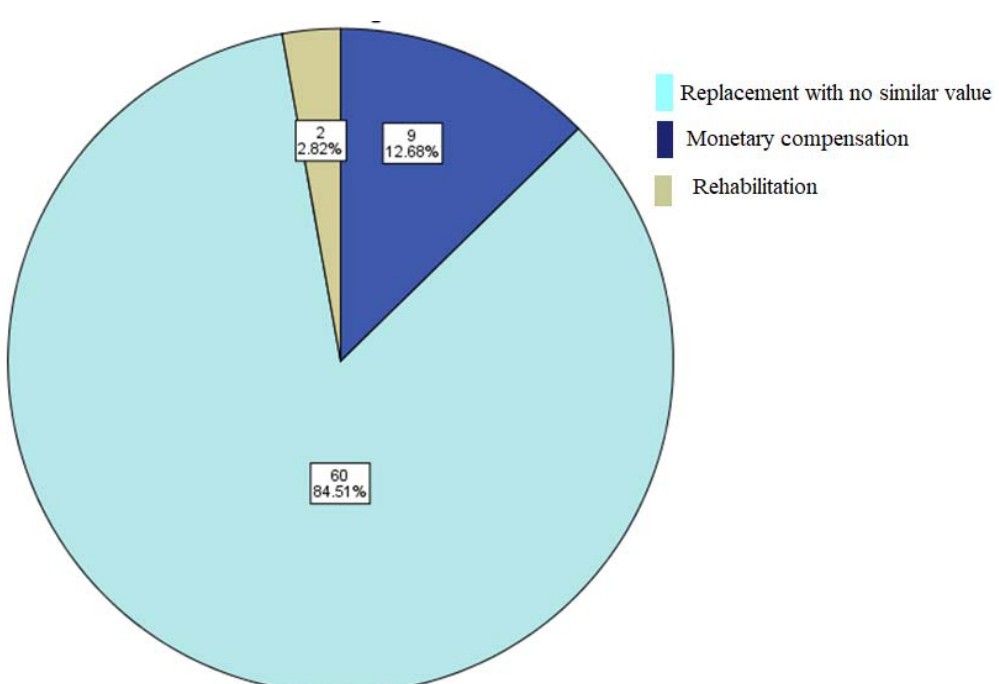

**Figure 6.** Enhancement and mitigation measures for the loss of farmland.

## 5. Discussion

This section discusses the stakeholder engagement process, FPIF implementation, how impacted and interested stakeholders' opinions were considered, how the land deal is transparent, how LSAI caused mass displacement, and why the promised compensation system was ignored. It also explains why LSAI performance and benefits were low in Ethiopia's Shashamane rural district of Oromia. In fact, international law recognizes governments' and corporations' responsibility to engage affected communities, particularly in accordance with the principle of "Free, Prior, and Informed Consent," as outlined in ILO Convention 169 and the United Nations Declaration on the Rights of Indigenous Peoples [40]. According to Articles 43, 44, and 92 of the FDRE Constitution, all people have the right to full consultation and expression of views, compensation, including relocation with adequate state assistance, and participation in national development, including, in particular, the right to be consulted about policies and projects affecting their community, and the development of their capacities and meeting their basic needs [41]. In the Environmental Impact Assessment Proclamation (Proc. no. 299/2002) [42] and Environmental Policy of Ethiopia (EPE, 1997) [43], public involvement is a mandatory legal prerequisite for the implementation of major development projects, programs, and plans. This proclamation and policy is a proactive tool and a backbone for harmonizing and integrating environmental, economic, cultural, and social considerations into a decision-making process in a manner that promotes sustainable development. Further, the concept of a "right" to participate in decisions affecting one is strongly mentioned in the literature [10,54,55]. Proposals for increased participation have also been directed toward overcoming the adverse effects of policy outcomes [56]. Furthermore, participation has several potentials in conflict resolution, as a strategic maneuver to accomplish other unstated or stated objectives, and improve information inputs into administrative decisions [57]. Others also argue that participation is important to obtain traditional knowledge that may be useful for decision-making, improvement to the project design, and other intangible and incidental factors involved in the process [58].

Others have also argued that it is essential to involve the public in identifying the problems and data that could be crucial to project success. Local expertise was beneficially helpful to the project's growth and viability [8]. Many large-scale projects have failed because they either failed to win public acceptance and support or neglected to consider

local and traditional factors [8]. There are numerous disadvantages associated with public participation—it is time-consuming and expensive, and a major obstacle to efficient functioning of private businesses [56]. Furthermore, large-scale crop production projects that involve sowing, transplanting, growing, and harvesting plant material, including food crops and export/industrial crops, are boldly acknowledged as involving public consultation [42,43]. Overall, in our stud we found evidence for a lack of incorporation and interaction with view of affected and interested stockholders in the consultation process. Additionally, the result suggests that the following factors were deficient: establishing areas of agreement, legitimizing proposals, ensuring greater acceptance and support, and providing a disagreement handling mechanism to reach a common position. Hence, we argue that the Elfora Shalo–Melega LSAI was inconclusive, unfair, and not open and transparent, and therefore lacks credible and relevant participation. As a result, it cannot create and build confidence and trust between the government, the proponent, and the local community; this kind of consultation is categorized to manipulation and therapy (nonparticipation) [53]. In fact, many people are outraged because they did not receive complete and timely information. Moreover, Guaraldo Choguill, (1996) and Nolte and Voget-Kleschin (2014) argue that a weak consultation process leads to a total lack of sense of ownership and project collapse.

Based on the result of the survey, 98% of rural households in the three kebeles (B/Dannaba Toga and D/Calalaqaa) were directly or indirectly affected by the LSAI proposal. Besides the entire effect of LSAI, a significant number of households were displaced from their locality. Moreover, 52.9% and 91.7% of rural households were displaced from their farmland and communal grazing land without adequate compensation for losses. This involuntary displacement of local people and evictions of land users have decreased their living standards and livelihoods, and negatively affect the enjoyment of human rights, including the right to life, the right to food, the right to housing, and the right to health, and the property right. Many large-scale projects have failed because they either did not gain public acceptance and support or did not take into account local and traditional factors [59,60]. Indigenous or local peoples should never be subject to expropriation without their agreement, and no relocation shall take place without the free, prior, and informed consent (FPIC) of the indigenous people concerned and after agreement on just and fair compensation and, whenever possible, with the option of return [40]. In Africa, for example, customary rights hold 80 percent of the land, but only 3 percent of that land is legally owned by communities [59]. Government organizations that promote investment have frequently allotted property for commercial use while ignoring local populations' land tenure rights [60–63]. Hence, the Ethiopian government, which has undertaken a program of land registration since 2003, has not benefited local communities in the Shashamane rural district because of a low implementation rate to lower-level administration units, i.e., the kebele. The main objective of the project is to address the problem of tenure security, reduce land disputes and litigation, bring empowerment, and increase investment in land [64]. Hence, the success of large investment projects, together with the welfare of nearby communities, is impacted when a project does not appropriately take into account community land rights and usage. Moreover, in this regard, several instruments of international human rights recognize a smallholder's right to land, and its indispensability to realize other human rights [55]. For instance, the FAO's Voluntary Guidelines call upon states to respect, protect, and fulfill the land rights of smallholders concerning the right to adequate food [65]. Loss of access to the commons undermines local community livelihoods unless there is compensation by using land of equivalent or superior quality [39].

Government-granted land-based concessions to firms in emerging markets are projected to be occupied in 93 percent of cases. Therefore, the recognition of collective tenure rights to the commons is a cornerstone of sustainable development and optimizing scarce resources [45,66]. All land in the nation, whether it is urban or rural, is declared to be state property and private ownership is prohibited by both the federal and regional constitutions and existing land regulations. Land users (cultivators and pastoralists) only have use

rights to the land under their control; they are not permitted to exchange, sell, or mortgage that land in any way. Moreover, our survey results reveal that 86.5% and 91% of rural households indicated that the LSAI affects, without adequate compensation, the aesthetic resources and the cultural, religious, and historical heritage assets, respectively. Moreover, about 55% were unaware of any action taken by the government to mitigate traditional cultural values, spiritual assets, and tourist attractions. A similar study also reveals that the Government of Ethiopia evicts smallholders for stated purposes of promoting private investments, including for the promotion of large-scale commercial agriculture and urbanization, without adequate due process of adequate compensation and law [35,45,47]. Similarly, at all three sites (kebele), our study suggest that the LSAI was creating a shortage of communal grain land and increased the intensity of land conflict, causing the deterioration of livelihoods. The finding is in line with the observation that the LSAI is causing the loss of farm and grazing land; cultural, religious, and historical heritage assets; and aesthetic resources [21,67,68].

Moreover, our findings suggest that the LSAI was causing health-related problems and communicable diseases such as malaria, schistosomiasis, and diarrhea. Other study also documented LSAI impact on child health related issues [69]. Dwivedi (2002) looks at development-induced displacement (such as LSLIs) in two ways. The first argument is that development-induced displacement is inevitable, and minimizing the effect of displacement is necessary. The second view sees displacement as the ultimate ugly face of development. Instead of improving people's wellbeing, development—via displacement—causes the disruption of their existing ways of life and the denial of property rights. Generally, without taking either side of these views, this study suggests that the LSAI has displaced rural households involuntarily in favor of developing large-scale crop production projects (rain-fed and irrigation) and without providing adequate resettlement areas with appropriate housing and services (water and sanitation), and productive (cultivating) and grazing land. Instances of unreasonable displacement, where households are obligated off their land without their consent and compensation, and most of the negative impacts of LSAIs on the displacement of the local community, have been widely reported [70]. Another study in semi-agro-pastoral areas of Ethiopia confirmed that LSAIs are causing household displacement [71]. Moreover, this study suggests that the LSAI has created unplanned human settlements and disturbed the standard of living. However, if the displacement in the future is inevitable, it should be implemented with community consultation and adequate land improvement strategies [71,72].

Our survey result indicated that 65.6% of respondents appealed to the local government to monitor and evaluate the performance of the LSAI, while 93% of the respondents expressed willingness to return the compensation if the government permits the return of the previously owned land. Moreover, this study suggests effective monitoring and evaluation facilitate early identification of implementation challenges, while also facilitating corrective action and keeping implementation on track. In fact, the Elfora Agro-Industries P.L.C. Shalo–Melega LSAI agricultural project did not conduct an EIA (environmental impact assessment) prior to implementing the Elfora Shalo–Melega LSAI in Shashamane rural district; as a result, the LSAI effects are not minimized and the positive impacts are not enhanced. In many cases, the implementation of investment projects begins before the EIA is submitted and approved [73]. This lays the foundation for future improvements [74]. Companies and investors who cannot recognize and engage effectively with local stakeholders may suffer significant financial, operational, and reputational risks [59]. Because conflicts can result in construction delays, business interruptions, compensation payments, or other indirect operating costs for businesses and investors, these risks are sometimes only noticeable to the firm management [75].

Hence, our closure examination shows that the issue of land dispute is common in the study area. To begin, there are four different types of land disputes that are pertinent to this study: disputes among farmers, disputes between farmers on the one hand and the government on the other, disputes between farmers on the one hand and investors on the

other, and disputes between an investor on the one hand and the government on the other. Over time, disputes between farmers and investors have become increasingly violent, and when this happens, the federal and regional security forces frequently step in to mediate the situation and prevent farm equipment, irrigation systems, and crops from being destroyed by enraged local farmers and landless youths. According to a Human Rights Watch (2016) report, one of the key causes of Ethiopia's 2016 government shift was the "Addis Ababa Integrated Development Master Plan," a controversial proposal to expand the municipal boundaries into the farmland of the Oromia region and the lack of monitoring, low gain, and corruption of megaprojects such as the LSAI. Shashamane's urban and rural districts were among the protest locations in 2016.

## 6. Conclusions

Significant findings are drawn from this study. When examining stakeholder consultation and local rural communities' involvement in the LSAI, it is clear that consultation with affected and interested parties was required for major projects such as large-scale rain-fed and irrigation projects. Among the importance of consultation and including the view of those affected and interested in the project, was also to address the negative potential social, economic, and environmental impacts of development projects. As a matter of fact, transparent land transactions must include community participation. In this regard, we argue that LSAI proponents and investors and the government should open their doors to involve local people and to implement FPIF, which must also constitute a conclusive, fair, open and transparent, credible consultation process. This builds confidence and trust among the government, the proponents, and the local community.

Public participation must at least give individuals who will be directly affected by the plan a chance to voice their opinions and express any potential social, environmental, or other effects. Concerning this, our study reveals that households that a community with an LSAI have reacted unfavorably to the LSAI projects from the very beginning, partly because they were neither consulted openly nor informed in the first place, and partly because of their fear that such an LSAI will have an unwelcome consequence on their settlement, farm grazing, and forest land. Additionally, this study offers proof of the LSLI-induced displacement and compensation mechanism in Ethiopia's Shashamane district. The findings show that the LSAI is forcing local rural households out of their settlement areas, leaving them without suitable housing and services (such as water and sanitation), and preventing them from accessing pastures and other resources in the research region. Ethiopia has always prioritized small-scale agricultural production as a development strategy, despite recently adopting a plan to promote LSAIs. Export-oriented agricultural investment is a key component of Ethiopia's overall development strategy, which calls for the country to reach middle-income status by 2025; in this regard, the LSAI was affecting local community residents, farms, and grazing land. If relocation is inescapable, it is best to obtain everyone's cooperation in advance, pay them, and give everyone access to communal resources.

Corrective action is also required to give the displaced local community access to resources from the common pool. Access to grazing areas can be guaranteed for rural people through responsible agricultural practices on LSLIs. More specifically, the government should: (1) closely monitor the proper implementation of investment projects for which small-scale farmers' land has been appropriated and (2) evaluate the processes and outcomes in terms of their potential benefits to the disadvantaged smallholders and rural people whose livelihoods solely depend on their lands. The study also revealed that land is the most important source of income, if not the only one. Furthermore, the region had few other sources of income and was dependent on smallholder agriculture. The published literature also implies that enhancing monitoring and evaluation closely improves the proper implementation of investment projects. (3) The government should also revise the compensation policy and resettlement policies, and the Ethiopian and Oromia Regional State Investment Authority should adopt guidelines and approaches that regulate LSAIs

to ensure the protection of the tenure systems and specifically consider land rights in the investment. Compensation should, as a matter of legitimacy, lead people to better lives. However, we contend that governmental protections of community rights must be respected and upheld. States can fulfill their obligations and fulfill their responsibilities by putting into practice concepts such as participation, FPIF, nondiscrimination, and accountability. In the event of opposing land claims, it is crucial to account for the land tenure system, current inequalities, and inequities while also providing an effective means of resolving conflicts.

**Author Contributions:** Supervision, D.T.; Writing—original draft, Y.A. All authors have read and agreed to the published version of the manuscript.

**Funding:** This research received no external funding.

**Institutional Review Board Statement:** Not applicable.

**Informed Consent Statement:** Not applicable.

**Data Availability Statement:** Not applicable.

**Acknowledgments:** We also value the unfettered help provided by the unnamed Shashamane rural district officials during the field work.

**Conflicts of Interest:** The authors declare no conflict of interest.

## Appendix A

**Table A1.** Sample households from a directly impacted population.

| District | Kebeles | Total Population | | Proportional to Size (PPS) Systematic Sampling Techniques | |
|---|---|---|---|---|---|
| | | **Male** | **Female** | **Total Population/HD/** | **Proportional to Size (PPS)** |
| Shasamane adjacent district (treatment) | B/Dannaba | 827 | 113 | 940 | $(940 \times 153)/2098 = 69$ |
| | Toga | 540 | 96 | 636 | $(636 \times 134)/2098 = 41$ |
| | D/Calalaqaa | 416 | 105 | 521 | $(521 \times 134)/2098 = 33$ |
| | | | | Ground total | 134 |

## Appendix B

**Table A2.** Performance, monitoring, and follow-up of the LSAI.

| Please Answer the Following Questions | | Frequency | Percentage (%) |
|---|---|---|---|
| Have you heard any measures taken by the government to monitor and evaluate the performance of LSAI including? | Yes = 1 | 88 | 65.6 |
| | No = 0 | 46 | 34.3 |
| Willing to return compensation | Yes = 1 | 125 | 93.2 |
| | No = 0 | 9 | 6.7 |

**Appendix C**

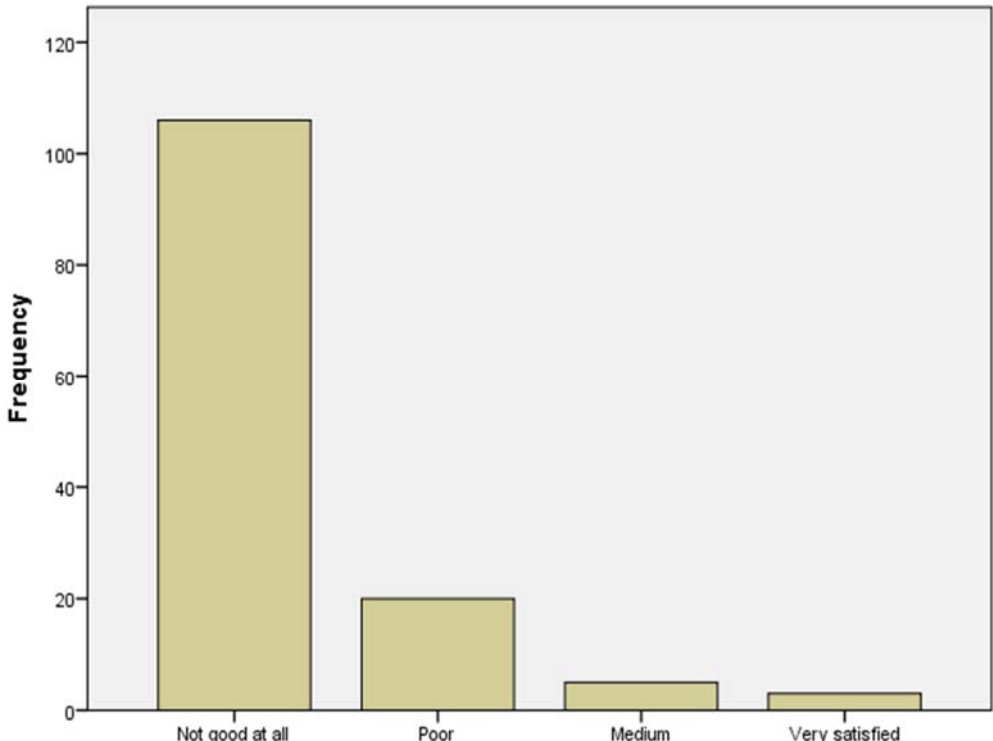

**Figure A1.** Local Community's satisfaction with the overall project.

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
