# Peer review of "Consultation and Displacement in Large-Scale Agriculture Investment: Evidence from Oromia Region’s Shashamane Rural District"

_land, doi:10.3390/land11091384_

Round 1

Reviewer 1 Report

Abstract:

1.      Please, review and shorten the following paragraph. There are too many details about the study area that shouldn't be included in an abstract: "Shashamane rural district was selected as a target area of Large-Scale Agriculture Investment (LSAI) to improve agricultural production and ensure local economic development by the government and private (domestic and foreign) investors. The area is rich with the availability of fertile soil, ground and surface water, vegetation, and a stable climate to grow different commercial crops for the domestic and export markets. Shallo-Melega farm project is one of the private LSAI projects that started operation in 2008 in the Shashamana rural district. This farm project comprises a crop production site, construction of a road, a crop storage facility, and developing irrigation from a total of about 24,710.51 acres of land along the central Rift Valley basin, for long-term leases. Little attention has been given to community consultation, Free, Prior and Informed Consent (FPIC), and local displacement. Henceforth, the study examined the consultation process, application of FPIC, and local community displacement due to LSAI in the Shashamane rural district."

2.      What is the main objective of your research?

3.      Furthermore, the detail "in the vicinity within a 10km radius" is too much for an abstract.

4.      A "who" is missing before "expressed" ("99.3% of respondents expressed")

5.      At first glance, the sentence 'Generally, the consultative process was manipulation and therapy' is confusing to the readership. The information in the abstract must be clear, pls. rephrase it.

Introduction:

6.      You have an Introduction section of one long (two pages) paragraph. Split it into at least 6-7 paragraphs.

7.      "Africa, it argues, is a prime target". Delete "it argues".

8.      You must update the information ((Deininger & Byerlee, 2011, Smaller & Mann, 2009; Anseeuw, 2013, Cotula et al., 2009; Dessalegn, 2011) on land grabbing with predominant new references (2019-2022 and even 2023).

9.      You said, "This study aimed to comprehend three questions against this background," and you included four questions. Are these four questions the research questions of your study? If yes, pls., rephrase them to sound like RQs.

10.   There is rich literature dedicated to LSAI in Africa. What is the contribution of this paper to the state-of-the-art?

Research methodology and research approaches

11.   A study area map would be handy for the readership (subsection Background overview study area and the project)

12.   Sometimes you wrote "Kebele" with a capital letter, other times with a lowercase letter. Pls, ensure uniformity across the ms. The same comment for "respondents".

13.   Correct the statement: "4-point scale (0 being not at all satisfied, 1 being poor, 2 being medium, and 5 being very satisfied)"

14.   More concretely, what was the objective of the survey?

15.   What do you mean by "314 (100%) of the 134 households that took part in the survey responded quickly to the questionnaire."?

16.   To better understand the research methodology, you must include a table with the variables investigated.

17.   For subsections B and C, you have to explain if and how you followed COREQ and give details about the three main domains of COREQ.

18.   What do you mean by "the research team also used photography in a safe, legal, and morally acceptable manner". This is  a very sensitive issue.

Results:

19.   A higher resolution for Fig 4 and 5

20.   This part belongs to the Discussion section, "It is essential to involve the public in identifying the problems and data that could be key to the project success. Local expertise may also be extremely helpful to the project's growth and viability. Many large-scale projects have failed because they either failed to win public acceptance and support or because they neglected to consider local and traditional factors."

Discussion:

21.   This section must be rewritten because it does not discuss the results of the study but refers to other aspects that rather belong to the Introduction: for example, "Hence, international law recognizes the responsibility of governments and corporations to engage affected communities, particularly with the principle of "Free, Prior, and Informed Consent," ..., including food crops and export/industrial crops are boldly recognized as undertaking public consultation(EIA, 2002; EPA, 1997; Land 2021, 10, x FOR PEER REVIEW 13 of 19 Wathern, 1990)" or "In Africa, for instance, 80% of the land is held through customary rights, but communities legally own just 3% of that land", etc

22.   Again, a single paragraph for an entire chapter. Split it into several paragraphs.

23.   This section must explicitly show if and how the results respond to the four questions mentioned in the Introduction.

24.   Conclusions:

25.   Pls, revise the English grammar and style for this section. It isn't easy to follow your ideas.

26.   It would be best if you split the Conclusions section into several paragraphs.

27.   Remove the citations from the Conclusions.

Miscellaneous: Extensive revision of the English grammar and style is required for the entire ms (e.g., "conceptual formwork", "The business models in place, are the culmination of")

Appendix 3: The resolution of the figure is not good. Pls, replace the figure.

Author Response

Dear Reviewer one,
We are very thankful for your positive comment and suggestion. We believe that
all the comments and suggestions are very useful to improve the entire paper. So,
according to your positive comments and suggestions we incorporate all
comments
and suggestions point by point. Now it is improved from the originally
submitted.

The authors

Reviewer 2 Report

This paper examines an important issue, but it needs significant improvements before it could be considered for publication.

To start, the paper is sprawling and needs to be better organized and written more clearly. Paragraphs are pages long and thus reduce the reader's ability to understand the main points expressed in the paper. This is a problem throughout the paper, starting from the abstract and introduction to the subsequent sections of the paper.

Next, the front end of the paper needs to more clearly articulate the existing conceptual, empirical, or methodological research gaps in the field and how the paper plans to fill these research gaps. Although the paper does have research questions at the end of the introduction, they are bit too technical and not related to gaps in the literature. They are also buried in an extremely long paragraph. The authors may also want to consider writing a shorter introduction and then separate the literature into a separate literature review section afterwards.

In the methods, the authors could explain in more detail why these methods are used: what is the broader study design? What are the strengths and weaknesses of using this set of methods?

In the results section, the authors could develop a means of more clearly showcasing the data. This means cleaner and simpler tables and graphics as well as description of what the findings show and analysis of their significance.

In the discussion and conclusion, the authors could more clearly articulate why the findings are important for existing research, future research, and policy.

Overall, this paper has some interesting findings, but they are obscured by the sprawling and long-winded paragraphs. The writing needs to be done in a clearer way. This includes shorter paragraphs, better organization, and clearer sentences.

Good luck with the revisions. 

Author Response

Dear Reviewer ,

We are very thankful for your positive comment and suggestion. We believe that all the comments and suggestions are very useful to improve the entire paper. So, according to your positive comments and suggestions we incorporate all comments and suggestions point by point. Now it is improved from the originally submitted.

The authors

Round 2

Reviewer 1 Report

The content of the ms improved from the first version. There are still several aspects that must be considered:

- A higher resolution for all figures is needed;

- Replace “CORE” with “COREQ”;

- A table with the variables investigated is still missing.

- Practically, did you perform a focus group or an interview group?

Author Response

Authors response  

Point 1- A higher resolution for all figures is needed;

Response 1: All figures resolution is improved from 600dti to 720 dti resolution

-Point 2:  Replace “CORE” with “COREQ”;

Response 2: Replaced

Point 3: A table with the variables investigated is still missing.

Response 3: Table is included (Table 2. List of major variables investigated).

Dimension

Variable name

Variable type

Unit of Measurement

Recognition and sense of ownership

Meetings

Dummy

1, if invited to a stakeholder meeting, 0 if  not  

Participation   

Dummy

1, if participating in the  proposal, 0 if  not 

Proposal explanation

Dummy

1, if the proposal is adequately  explained to  all stakeholders, 0 if  not 

effects and hazards

Dummy

1, if the proposal effect and hazards are adequately  explained, 0 if  not 

Power and level  of influence

Responsibility

Dummy

1, if the local community is influenced and empowered, 0 if  not 

Decisions

 Dummy

1, if the decision of the local community incorporated, 0 if  not 

Degree of community control

Dummy

1, if higher degree of community control or partnership, 0 if  not 

Key principle of participation

Open and transparent

Dummy

1, if  participation is Open and transparent, and understood, 0 if not

Fair and neutral,

Dummy

1 if, the consultation is  fair and neural, 0 if not

Inclusive

Dummy

1, if the proposal is  inclusive, 0 if  not 

Relevant

Dummy

1, if the proposal is, 0 if  not 

Responsive

Dummy

1, if  the proposal is responsive to stakeholder input, 0 if not

Credible

Dummy

1, if the proposal is credible, 0 if  Otherwise

Impact

Direct impact

Dummy

1, if the proposal is a direct impact, 0 if  not 

Displacement e resident

Dummy

1, if the proposal is to displace you from your residential, 0 if  not 

Displacement from  communal land

Dummy

1, if the displace your from communal land, 0 if  not 

Indirect impact

Dummy

1, if the proposal is an indirect impact, 0 if  not 

Cultural sites

Dummy

1, if the proposal is adequately  explained, 0 if  not 

Heritage

Dummy

1, if the proposal is affecting heritage  explained, 0 if  not 

Biodiversity

Dummy

1, if the proposal affects biodiversity, 0 if  not 

Mitigation and compensation 

In-kind compensation

Dummy

1, if in-kind compensation is provided, 0 if Otherwise

Monetary compensation

Dummy

1, if the monetary compensation is provided, 0 if  not 

Resettlement

Dummy

1, if  compensation is provided resettlement, 0 if Otherwise

Site remediation

Dummy

1, if  Site remediation is provided, 0 if  Otherwise

Mentoring

Monitoring and follow-up

Dummy

1, if the monitoring  and follow-up are adequate, 0 if  not 

Satisfaction 

Satisfaction

Rank (4-point scale)

0 being not at all satisfied, 1 being poor satisfaction, 2 being medium satisfaction, and 3 being very satisfied)

Point 4- Practically, did you perform a focus group or an interview group?

Response 4: Yes, we conducted 6 FGD, 2 FGD per Kebele (kebele lower administration unity, in Ethiopia). Moreover, tools and guides were used during Focus-Group (FG)

Kebele name

Number of people involved

Phase

Toga

6

1

Toga

7

2

B/Dannaba,.

10

1

D/Calalaqaa

6

2

D/Calalaqaa

7

1

6

2

Reviewer 2 Report

The authors have responded adequately to the reviewers' comments. This paper is significantly improved. This paper will be ready for publication once it has been read through one more time for clarity and flow.

Author Response

Point 1: The authors have responded adequately to the reviewers' comments. This paper is significantly improved. This paper will be ready for publication once it has been read through one more time for clarity and flow.

Response 1: The authors reviewed the manuscript several times to enhance its coherence and clarity.